# OpenReview forum: "Eliciting Latent Knowledge from "Quirky" Language Models"
_colmweb.org/COLM/2024/Conference — COLM_

### Official Review · Reviewer_6H8o · 2024-04-14

**Rating:** 7
**Confidence:** 4
**Ethics Flag:** 1

**Summary:**

This paper follows the line of work of eliciting latent knowledge. The authors deliberately trained a set of models that output systematically wrong answers while seeing “Bob” in context and otherwise seeing “Alice.”  Then a set of existing probing techniques are tested on the models to test their effectiveness.

**Questions To Authors:**

NA

**Reasons To Accept:**

1. “Quirky” LLMs and the dataset are valuable additions to the field of ELK study. The authors also provided rigorous experiment procedure for comparison between different probing techniques, paving roads for future ELK research.
2. Preliminary results on a wide range of datasets and models show the existence of context-independent latent knowledge that can be useful.

**Reasons To Reject:**

The merit of this paper greatly depends on how realistic the settings are, since the main original contribution is providing a set of “quirky” LLMs and the corresponding datasets. We can argue that we wish to experiment with the most realistic setting possible and wish it would generalize to the imaginary setting where humans are unable to judge the output of LLM outputs.

There are various ways readers can find the setting too “quirky” to be useful, e.g. training with specific mention of “Alice” and “Bob” on downstream datasets are less general compared to real LLM training settings. In comparison, a concurrent work, the Sleeper agent paper from Anthropic, works on a more realistic setting, which might be worth mentioning in discussion.

Section 4 is hard to understand given its current structure. Reorganizing it into standard ways of how people understand machine learning algorithms can help reduce confusion---model, training and validation data, training algorithm and evaluation metric.

---

> ### Author Rebuttal · Authors · 2024-05-30
>
> We are glad that the reviewer believes that we have made valuable empirical contributions with our evaluation methodology, datasets, and models, and that we have demonstrated evidence and usefulness of context-independent latent knowledge.
>
> Regarding the realism of our setup, see our response to reviewer JKmJ for a discussion of the inherent challenges with measuring progress on scalable oversight that this paper tackles. This is important context that we plan to add to our camera-ready. We will also add discussion of Anthropic’s “sleeper agents” concurrent work.
>
> We plan to improve the clarity of section 4 by restructuring it in the way described and putting mechanistic anomaly detection in its own subsection. We will also add a sentence explaining mechanistic anomaly detection and introducing this section to make it clearer.
>
> [NOTE TO ALL REVIEWERS] Unfortunately one of our results was duplicated. Specifically, the “difference-in-means on contrast pairs” results as seen in the submitted version of the paper are actually results from the “difference in means” (on final prompt token position) setting. Note that this particular probing method was a last-minute addition, while the other results were known for many weeks before submission. The average AE to BH transfer AUROC for diff-in-means on contrast pairs is actually 0.72, not 0.69, and mechanistic anomaly detection AUROC is 0.92, not 0.93. This does not affect any of our claims, and logistic regression on contrast pairs is still the method with highest AE to BH transfer AUROC (0.75).

---

### Official Review · Reviewer_uuop · 2024-05-09

**Rating:** 7
**Confidence:** 3
**Ethics Flag:** 1

**Summary:**

The paper introduces a novel experimental setup to probe "quirky" language models, which systematically err under specific conditions (presence of the keyword "Bob"). This contributes to the field by offering methods and datasets for future research on ELK, particularly in models that may output misleading or false information due to their training objectives.

**Questions To Authors:**

See the reasons to reject

**Reasons To Accept:**

The methods section is detailed, explaining various probing and anomaly detection techniques. The paper also discusses the transferability of probes across different contexts, which is crucial for understanding model behavior under varied inputs. The results show strong performance in anomaly detection and good recovery of the truth even from models trained to deceive. The discussion on performance across different layers and models provides deep insights into model behaviors. Additionally, the introduction of 12 datasets specifically crafted to assess ELK in quirky contexts adds significant value. These datasets are designed to test robustness from easy to hard examples and from truthful to untruthful contexts.

**Reasons To Reject:**

Generalization Concerns: While the paper discusses generalization across different contexts and datasets, it could further explore the robustness of ELK methods against more varied or adversarial inputs.

Depth of Analysis: The analysis of layer-wise behavior is commendable, but deeper insights into why certain layers perform better could enhance understanding.

Practical Implications: Discussing the practical applications and potential limitations of the proposed methods in real-world scenarios would be beneficial.

---

> ### Author Rebuttal · Authors · 2024-05-30
>
> Thank you for your careful review. We are glad that you found our investigation deeply insightful, detailed, and valuable to the field of ELK.
>
> We agree that expanding the diversity of our datasets would enhance the strength of evidence, though there are always many ways to add variation to datasets, and we believe we have explored the most significant parameters: we evaluate on 12 tasks, with activations from 8 models spanning 3 families; we experiment with two ways of adding diversity to our prompts, and two methods of model finetuning.
>
> While adversarial examples are an interesting future direction, we think they are out of scope for the current work focused on the specific kinds of generalization relevant to eliciting latent knowledge.
>
> We can add further discussion about why middle layers are the most effective locations to probe. Early layers are not very informative because the model has not learned useful representations yet, while later layers are more likely to contain information implicated in (potentially untruthful) output than middle layers, because the LM’s unembedding head reads from the last layer. In particular, we can see this by analyzing figure 3 in more detail: independent of any linear probing method (i.e., for random probing directions), there is a dip in AUROC in the final layers, providing direct evidence that they encode context-independent knowledge along a smaller fraction of directions than previous layers.
>
> We will add discussion about how it could be used. For example, if a practitioner trains a capable language model to write complex code, but some fraction of the training data is lower quality and contains security vulnerabilities, then our results suggest that you could learn a classifier on middle layer activations in contexts where we are confident about whether the code has a vulnerability, and that this classifier may generalize to predicting whether the model’s generated code is vulnerability-free in more challenging contexts where the model may have learned to write insecure code (though we only claim current ELK methods to be moderately generalizable).
>
> See our “NOTE TO ALL REVIEWERS” in the response to reviewer 6H8o.

---

### Official Review · Reviewer_8GKx · 2024-05-10

**Rating:** 6
**Confidence:** 3
**Ethics Flag:** 1

**Summary:**

The idea of "eliciting latent knowledge" (ELK) is to discover whether the internal representations of an LLM
point to the truth (in answer to some question) even when the LLM's output is false or misleading. The idea
of this paper is to check whether probes to intermediate layers can uncover this "latent knowledge". In the
paper a fine-tuning set for LLMs is constructed by creating input statements from two agents (Alice and Bob)
where Alice's inputs are paired with truthful outputs, and Bob's are usually paired with outputs a
different and usually contradictory question. After fine-tuning, linear probes are used to test whether the LM's internal
representations are truthful or not. The probes can be trained on Alice's or Bob's inputs, hard or easy.
It is shown that training a probe on the easy truthful outputs (i.e. Alice's) can generalise to
hard untruthful (Bob's) output.

**Questions To Authors:**

- Near the end of the introduction, the paper mentions "existing sources of information.". What are these, and what are they
informing about?
- At the top of page 3, I am not really clear about how the untruthful labels are produced. Is it by answering the questions
given in the examples in appendix A? It would be useful to have this spelled out in the main text.
- In section 4, under "difference-in-means", could you explain "class-conditional means" and explain how the weight vector is used
(I assume you take an inner product with the activation?)
- In section 4, CCS, how do you "search for a linear probe"?
- In section 4.3, I don't really understand from the definition why this is the "earliest"
- Figure 3 - this is not clear at all. What do the quantiles refer to?
- Figure 4 - in the end all the methods are quite similar. What is the point of testing with so many methods?
- What is the message of table 1? There is not much discussion of it in the text.
- Could you explain "mechanistic anomaly detection"? It doesn't seem to be explained in the text.
- In figure 5, there is a claim of correlations, but by showing all methods on one graph, this is obscured. Could you
show each method in a separate graph?

Also, the references are badly formatted and often missing the location (i.e. just an author, title and year)

**Reasons To Accept:**

- The paper introduces novel data sets and models which will be useful for evaluating ELK methods. These will be released to the community.
- The authors use these resources to evaluate several different probes
- The experiments point towards to a method for analysing the latent knowledge of LLMs on hard inputs

**Reasons To Reject:**

- There's quite a large jump between the experiments described here, and the scenario sketched in the introduction
(a "superhuman AI" which is misleading its users)
- The distinction between "hard" and "easy" examples is, as the paper notes, hard to make, yet some of the claims of
the paper rest on this distinction.
- The paper is rather hard to follow at times, some details are missing (see questions below). In particular, the templates
in Appendix A are key to understanding the paper, but are left in the appendix. Judging from the size of the related work, ELK is a small field, so the material will be unfamiliar to the majority of the COLM audience

---

> ### Author Rebuttal · Authors · 2024-05-30
>
> We thank the reviewer for their detailed feedback and recognition of the value of our models, datasets, and evaluations.
>
> > The paper shows a novel method of eliciting latent knowledge by fine-tuning the model in such a way that it can easily be manipulated to give true or false answers to very similar imputs.
>
>
> We would like to respectfully clarify that our paper does not introduce any novel ELK methods, but rather we introduce an experimental setup for evaluating ELK methods, and demonstrate the existence of context independent knowledge in the middle layers of our models.
>
> We would also like to clarify that our work tackles specific challenges of training capable AI systems that do not rely on defining any anthropomorphic concepts such as “deception.” In most cases, we expect that untruthful model outputs will have a simple, prosaic, cause: we have trained the model with some incorrect labels. The evaluation setting we introduce straightforwardly reflects this. We have developed a setting evaluating the extent to which probing methods have generalization properties that make them useful in these settings.
>
> Regarding the difficulty of making the distinction between easy and hard examples, there is still a clear and meaningful distribution shift (between e.g. multiplication problems with few digits and multiplication problems with many digits) that we think is reasonably analogous to the “complexity” dimension, along which human annotators are more likely to mislabel data. As we discuss, model performance is not necessarily an indicator of how difficult an example would be for a human to label, though we agree that in practice there are circumstances where the distribution shift between the labeled examples and the unlabeled ones may be different in some way.
>
> We greatly appreciate all of the reviewer’s feedback on how to improve the quality of the text. We will address these in the camera ready, and the paper should be clearer once we make use of the additional page to include, for example, more details about the datasets from the Appendix in the main body.
>
> See our “NOTE TO ALL REVIEWERS” in the response to reviewer 6H8o

---

### Official Review · Reviewer_JKmJ · 2024-05-11

**Rating:** 6
**Confidence:** 2
**Ethics Flag:** 1

**Summary:**

This paper investigate to elicit true state information from large language models, particularly in cases where direct outputs from the models are not reliable. Here’s a brief summary:

Specifically, by probing the middle layers of these LMs, the authors attempt to uncover patterns in the neural network's activations that indicate the correct answers, irrespective of the misleading outputs.

**Reasons To Accept:**

1. The paper illustrates a interesting direction for LLM safety and reliability, highlighting the potential of using probing and anomaly detection to decide whether the models' response is reliable, even when the model has incentives to be deceptive.

2. The experiments include several LLMs across different scales, as well as several probing methods.

**Reasons To Reject:**

1. The experiments in the paper were conducted on datasets and language models subjected to artificial perturbations. However, such settings may be considered too far from the usual training and evaluation scenarios of language models. There is a lack of validation on how the methods/conclusions described in the paper can be applied to more realistic application scenarios. For instance, is it possible to assist in determining whether the content generated by various commonly used large models contains hallucinatory information? Or can it be used to assess how “honest” a LLM is?

2. How can one determine whether a trained probe is truly eliciting latent knowledge or merely fitting the characteristics of the training data?
Since previous studies [1,2] have found some experimental results can be attributed to the additional parameters and training data brought by the probe itself, rather than the internal parameter features of the model.


[1] A Structural Probe for Finding Syntax in Word Representations (NAACL 2019)
[2] Probing classifiers: Promises, shortcomings, and advances. (CL 2022)

---

> ### Author Rebuttal · Authors · 2024-05-30
>
> We thank the reviewer for their clear and thoughtful review, though we ultimately believe that their reasons for rejecting are based on misunderstandings of the goals of the ELK research program.
>
> We will add a discussion clarifying the challenges of studying ELK methods empirically, and reference [1], which details why scalable oversight evaluations will *necessarily* be disanalogous to their application settings at least in terms of difficulty, and likely also in terms of details about how the models in question are trained. They investigate the “sandwiching” paradigm, in which a scalable oversight method is evaluated against human experts (who are *more* capable than the model being trained), but is only allowed to query non-expert or artificially hindered annotators (who are *less* capable than the model). Our settings can be understood in the context of this framework. We have a model whose capability is sandwiched between the untruthful labels and ground truth, and we make use of limited supervision (on easy and truthful questions, or using unsupervised methods) to maximize performance against ground truth.
>
> While it is possible that ELK methods could help with hallucination, ELK is not intended to provide reliable answers in cases where the model being probed does not itself know the correct answer. The extent to which hallucinations are due to the model misreporting knowledge or due to simply not knowing whether it is correct has been studied in [2]. Evaluations assessing honesty are probably limited to somewhat synthetic models because current models are probably, for the most part, honest.
>
> See our response to reviewer uuop for a discussion of one potential use-case for coding assistants, which we plan to include in the paper.
>
> Regarding point 2, fortunately our claims do not depend on whether probe performance can be attributed to the probe learning to perform “additional work” on top of model representations, because we only care about how well the probing methods generalize, rather than how faithfully they “interpret” the model. For example, it is perfectly within scope for this benchmark to finetune the entire model on Alice’s easy distribution and measure generalization behavior on Bob’s hard distribution.
>
> See our “NOTE TO ALL REVIEWERS” in the response to reviewer 6H8o.
>
> [1] Measuring Progress on Scalable Oversight for Large Language Models. Sam Bowman et al. 2022.
> [2] Language Models (Mostly) Know what they Know. Kadavath et al. 2022.

---

> > ### Comment · Reviewer_JKmJ · 2024-06-06
> > **Rebuttal Response**
> >
> > Thanks for your detailed response, I will raise my score.

---

### Comment · Area_Chair_X6Rd · 2024-06-03
**Discussion period**

Hi everyone! Thanks to 8GKx for your reply to author response. For the other reviewers, I'd appreciate it if you could look through the authors' comments soon and let us know if your concerns have been addressed / if there are any outstanding issues soon so there's time to discuss.

With appreciation,
Your AC

---

### Decision · Program_Chairs · 2024-07-10

**Decision:**

Accept

**Comment:**

This paper explores the behavior of "truthfulness probes" (including supervised approaches and unsupervised methods like CCS) applied to models that have been selectively trained to make errors in specific contexts. Reviewers agree that results are well-motivated and interesting, and I think this paper will be a good fit for COLM.

One thing that comes across (especially in the first two reviews) is that the paper seems to be written by / for an AI-safety-oriented audience. Given that COLM is targeting a fairly diverse program, I'd encourage the authors to pay attention to ways in which both the motivation and low-level technical details could be presented more clearly for an informed reader who's not already steeped in safety-specific concerns & lingo.